# Effect of Polyphenols on the Ice-Nucleation Activity of Ultrafine Bubbles

**DOI:** 10.3390/nano13010205

**Published:** 2023-01-02

**Authors:** Tsutomu Uchida, Yukiharu Fukushi

**Affiliations:** 1Faculty of Engineering, Hokkaido University, Sapporo 060-8628, Japan; 2Faculty of Agriculture, Hokkaido University, Sapporo 060-8589, Japan

**Keywords:** ice, nucleation, polyphenol, freezing curve, dynamic light scattering, transmission electron microscopy

## Abstract

Ultrafine bubbles (UFBs) in water provide a large amount of gas and a large gas–liquid interfacial area, and can release energy through their collapse. Such features may promote ice nucleation. Here, we examined the nucleation of ice in solutions containing polyphenols and UFBs. To reduce the likelihood of nucleation occurring on the container walls over that in previous studies, we used a much larger sample volume of 1 mL. In our experiments, UFBs (when present) had a number concentration of 10^8^ mL^−1^. We quantified changes to the nucleation activity by examining the shift in the cumulative freezing (nucleation) probability distribution. Compared to pure water, this freezing curve shifts approximately 0.6 °C higher with the UFBs. Then, to the water, we added three polyphenols (tannic acid TA, tea catechin TC, and oligonol OLG), chosen because they had been reported to reduce the ice-nucleation activity of heterogeneous ice nuclei (e.g., AgI). We found experimentally that, without UFBs, all polyphenols instead shift the pure-water freezing curve to a higher temperature. Then, when UFBs are added, the additional temperature shift in the freezing curve is slightly higher for OLG, essentially unchanged for TA, and slightly lower for TC. To help to explain these differences, we examined the UFB size distributions using dynamic light scattering and freeze-fractured replicas with transmission electron microscopy, finding that OLG and TC alter the UFBs, but that TA does not.

## 1. Introduction

In nature, intracellular freezing is lethal to both animals and plants. As plants, unlike animals, cannot escape from cold conditions, they have various freeze-resistance strategies. For example, many plants living in mid- to high-latitude regions prevent intracellular freezing by promoting freezing outside the cells or organs. Another strategy is to accumulate freeze inhibitors such as polyphenols, sugars, and anti-freeze proteins. Such compounds promote the ability to supercool and avoid freezing [1,2,3]. 

In industry, freeze control can be useful. For example, in order to maintain food’s freshness while maintaining its quality, the food should be stored at the lowest possible temperature without being completely frozen. Recently, due to concerns about adverse health and environmental impacts from added compounds, methods involving natural additives that prevent freezing have been developed [4,5,6,7]. Similarly, the production of frozen foods, freeze-dried foods, and freeze-concentrated foods requires economical methods to control freezing [8]. 

On the other hand, water readily supercools. For example, the freezing of supercooled water droplets in clouds is important for generating rainfall as well as snowfall [9]. Hence, methods used to promote the freezing of supercooled droplets have long been studied. A common approach is to develop additives, such as silver iodide (AgI), to nucleate ice [4,5,6,7,9]. Unfortunately, most such additives adversely affect the environment, so research on more environmentally safe additives has continued. 

Sub-micrometer bubbles, known as ultrafine bubbles (UFBs) [10], have come under recent interest for various types of water treatment [8,11,12,13,14], and have been found to also promote the nucleation of the ice-like gas hydrates [15,16,17,18,19]. In the latter application, Uchida et al. [15] found that the dissociation of gas hydrates generates a large amount of UFBs that, upon later investigation, were found to promote the nucleation of gas hydrates [16,17,18,19]. Although UFBs were thought to rapidly collapse before they could affect crystal nucleation [20,21], Uchida and co-authors verified that they remain in the water long enough to promote nucleation [15,16,17,18,19].

Since the gas-hydrate crystals have an H_2_O framework similar to ice, we expected that UFBs would also promote the freezing of water. In terms of a freezing mechanism, the collapse of UFBs may generate radicals [12,13], a process that may nucleate ice in supercooled water; that is, UFBs may be a new material for promoting freezing without additives. 

In this paper, we first examined whether UFB-included water promotes ice nucleation, then measured the freezing curves of three polyphenols (natural additives), and finally investigated how the UFBs influence the freezing curves of the polyphenols.

## 2. Materials and Methods

### 2.1. Preparation of UFB-Included Water

UFB-included water was prepared using a commercially available UFB generator (Aura Tec, Fukuoka, Japan, type OM4-MDG-045) with 1 L pure water (ion-exchanged distilled water of resistivity of approximately 15 MΩ cm) and 0.25 MPa air at 293 K by immersing the water-filled beaker into the temperature-controlled bath (Otsuka Electronics, Osaka, Japan, type NM-454L) and running the generator for 1 h. This procedure produced UFBs in the bulk liquid phase (“bulk UFBs” or simply “UFBs” in this paper) with diameters of 100–500 nm and a number density of order 10^8^ mL^−1^. These bulk UFBs stably existed for approximately 1 week at room temperature [15,16,17].

After this generation process, some microbubbles (diameters between approximately 1 μm and 1 mm) also remained for a while (the turbidity of water disappeared within 15 min at room temperature). In addition, UFBs on the solid surfaces (called “surface UFBs”) may form on the reactor wall due to the high air supersaturation [22]. To help ensure that only bulk UFBs are present, we sealed the UFB-included water in a 200 mL glass bottle and stored it overnight at room temperature before use in the experiment. UFB number density and particle size distribution were hardly affected by this operation [14,15,16,17].

### 2.2. Freezing Assay of UFB-Included Water

We put 1 mL of either pure water (control) or UFB-included water in a reactor (1.5 mL microtube). On the outside of each reactor, a 0.1 mm ultra-thin thermocouple (T) measured the sample temperature. The reactor was then placed in a glass container to reduce heat transfer between neighboring samples. Approximately 20 reactors were set simultaneously in an insulated box in which each one was thermally insulated from its neighbors, and the box was put in a freezer at −16 ± 1.5 °C. This system allowed us to slowly and uniformly cool the samples so as to measure many freezing temperatures simultaneously under the same conditions.

By monitoring the sample temperatures with a data logger (Graphtec, Yokohama, Kanagawa, Japan: type GL220), we measured the time and freezing temperature every 10 s while the samples cooled from room temperature. As shown in Figure 1, the freezing temperature is the lowest temperature at the time when a sudden temperature rise occurs due to freezing (referred to as the “nucleation point”). The uncertainty in this temperature is ±0.5 °C.

We ran more than 50 tests under each condition, and measured the freezing temperatures. As freezing is stochastic, we used these data to construct the cumulative freezing (nucleation) probability distribution, hereafter just “freezing curve”. The distribution of a given preparation was compared to the control (pure water) case as follows. We plotted both freezing curves and calculated the area between the curves. If the distribution for the prepared case is shifted to a higher temperature than the reference, the area, hereafter the freeze accelerate coefficient (FAC), is positive; otherwise, the FAC is negative. Thus, if FAC is positive, the preparation has the effect of “promoting nucleation”, and if it is negative, it is evaluated as having the effect of “suppressing nucleation”. The uncertainty of FAC is estimated to be similar to that of the temperature measurements, at approximately ±0.5 °C.

### 2.3. Freezing of Polyphenol Solutions

Several polyphenols have been reported to suppress ice nucleation (or a supercooling-promoting activity) when used with ice-nucleating promoters (ice nuclei) [6,7]. Here, we examined three of these previously studied polyphenols to see if they have a similar effect on the freezing curve of UFBs. These compounds were tannic acid TA (hydrolysable tannin extracted from plants, Wako Pure Chemical Industries, Japan), tea catechin TC (flavanol extracted from plants, with purity of more than 40% [7], Genryoya, Takatsuki, Osaka, Japan), and oligonol OLG (condensed tannin with purity of approximately 40%, artificially synthesized compounds with a molecular weight distributed between approximately 290 and 4000 [7], Amino Up Co., Ltd., Sapporo, Hokkaido, Japan). These sources and purities were the same as those in the previous studies [6,7]. Each was dissolved in either pure water or UFB-included solution to concentrations of 0.1, 0.05, and 0.01 wt%. At these concentrations, the melting points of polyphenol solutions are estimated to be less than 0.02 °C [7], which is negligible compared to the temperature measurement accuracy. These aqueous solutions were used for the freezing experiments by the same freezing assay as described above.

### 2.4. Measurement of Interactions between UFB and Polyphenols

To clarify how UFBs interact with polyphenols, changes in the particle size distribution of UFB were measured using dynamic light scattering (DLS). The sample size in each measurement was approximately 1 mL. Light from a semi-conductor laser (70 mW, λ = 660 nm) was focused onto the center of the sample cell. The time-dependent intensity of the mixing of scattered light through two pinholes was then measured with the photomultiplier of the DLS spectrophotometer (nanoSAQLA, Otsuka Electronics, Osaka, Japan) at room temperature. The size distribution was computed with the Marquardt method. 

In addition, the states of UFBs in each aqueous solution were observed using transmission electron microscopy (TEM) on freeze-fracture replicas (FFT method) [14]. This electron microscope observation method helped us to infer the possibility of interactions between polyphenols and UFBs. For the FFT method, we prepared the samples as follows. A small amount (approximately 15 μL) of each solution was quenched in liquid nitrogen and fractured by a knife edge in a vacuum chamber at a temperature of approximately −150 °C and a pressure of approximately 10^−5^ Pa (JFD-9010, JEOL, Tokyo, Japan). To analyze the form of bubbles or aggregation phases in the solution, we made a replica of the fractured ice cleavage section by depositing platinum and carbon. Deposition was performed from an oblique direction on the section. The replica film was then taken off the ice sample by melting. The replica film was transcribed to a TEM-observation copper-grid (type F-400, Nisshin EM, Tokyo, Japan, opening size of 43 μm square) for observations with the TEM (JEM-2010, JEOL).

## 3. Results

### 3.1. Promoting Effect on Freezing of UFB-Included Water

The freezing curve in Figure 2 shows that the pure water (blue curve) froze in the range of −17 to −6 °C. This range is slightly warmer than that observed in previous studies [4,5,6,7], which may be due to our use of a larger sample. The UFB-included water case, plotted in red, froze over nearly the same temperature range, but shifted to slightly warmer temperatures. The resulting FAC value is +0.6 °C, indicating that the UFBs provide a slight freezing-promotion effect. In addition, this value (+0.6 °C) is close to the shift in the curves at the probability value of 0.5, which agrees with the value found in previous work, where it is defined as the FT50 parameter [4,5,6,7]. Therefore, the FAC value is consistent with the FT50 value, which is useful for evaluating the freezing behavior.

Assuming that the temperature dependence of a wall-induced nucleation process would probably differ from that of bulk nucleation, we suggest that the similar shapes of the two curves may indicate a negligible influence from surface UFBs on the freezing curve.

### 3.2. Effect of Adding Polyphenols on Freezing Temperature

We now examined the effect of polyphenols on freezing by comparing the freezing curves to that of pure water. As with the previous UFB case, the effect on freezing was quantified using the FAC.

For the case of TA, the freezing curve shifts to a higher temperature and sharpens. Specifically, for all concentrations, freezing occurs in the range of −13 to −8 °C (Figure 3). For the concentrations of 0.1, 0.05, and 0.01 wt%, the FAC values are +2.7, +2.8, and +3.0 °C, showing that the solution with the lowest concentration had a slightly larger nucleation-promoting effect.

A similar shift in the freezing curve occurs for TC solutions, except the distribution is even sharper. Specifically, freezing occurs in the range of −10 to −7 °C at all concentrations (Figure 4). The resulting FAC values are +3.5, +3.6, and +3.6 °C, again with the lower concentrations having a slightly larger promotion effect. 

For OLG solutions, freezing occurs within −12 to −7 °C for all concentrations (Figure 5). Thus, OLG also promotes nucleation and has a narrower freezing distribution over that of pure water. The resulting FAC values are +3.3, +3.3, and +3.2 °C in descending order of concentration. These values are intermediate between those of TA and TC, and like these other preparations, the concentration dependence is small.

These results show that all three types of polyphenol solutions exhibit a clear nucleation-promoting effect. For the concentration of 0.1 wt%, the FAC values of the three types of polyphenol solutions are +2.7 to +3.5 °C. However, for the concentration range measured (0.01 to 0.1 wt%), none show a clear dependence of the FAC on the polyphenol concentration, although, for TA and TC, the lower concentrations have slightly higher FAC values.

Compared to previous studies, we used a larger amount of aqueous solution. For the case of pure water, we found a lower amount of supercooling than that in a previous study [5]. With the larger amount of water, we also expected the system to exhibit either the nucleation-promoting or the nucleation-inhibiting effect relatively clearly. Although these polyphenol solutions suppress ice nucleation when used with ice-nucleating promoters (ice nuclei) [6,7], their airborne ice-nucleation properties (with no ice-nuclei additives) are small and show variable behavior [5,6]. Compared with previous work [6], the nucleation-promoting effect of TC was qualitatively consistent, but the results of TA and OLG differed. The reason for the discrepancy is not clear, but the effect of disturbance in the aqueous solution also increased with a larger volume of aqueous solution, and the additive may have further increased the probability of heterogeneous nucleation. 

### 3.3. Effect of UFBs on the Nucleation Promotion from Polyphenols

We now examined how adding UFBs to the polyphenol solutions affects the freezing curves. For these experiments, we used a 0.1 wt% concentration of each polyphenol solution and quantified the effect using their FAC values (with respect to pure water and also to other preparations). 

The UFB-included solution with 0.1 wt% TA (blue solid triangles in Figure 6) promotes nucleation more than that of an aqueous solution including only UFBs (red solid line). In particular, the FAC value with respect to pure water is +2.8 °C, whereas that with respect to UFB-included water is +2.2 °C. The FAC values are the same as those with only the TA 0.1 wt% aqueous solution; that is, the presence of UFB in the TA solution has almost no effect on the freezing activity.

For the corresponding TC preparation, TC with UFBs (green solid triangles in Figure 7) also promote freezing compared to the sample with only UFBs (red solid line). The resulting FAC value with respect to pure water is +3.3 °C, whereas that with respect to UFB-included water is +2.7 °C. In this case, the presence of UFBs suppresses the nucleation-promoting effect of TC by approximately −0.2 °C. As this value is less than the temperature uncertainty, the influence of the UFBs may be statistically insignificant.

For the OLG preparation, the UFB-included aqueous solution with 0.1 wt% OLG (red triangle, Figure 8) also promotes freezing compared to the sample with only UFBs (red solid line). The FAC value with respect to pure water is +3.5 °C, whereas that with respect to UFB-included water is +2.9 °C. The FAC with respect to the OLG 0.1 wt% aqueous solution is +0.3 °C (i.e., without the UFBs). As this value is less than the temperature uncertainty, the influence of the UFBs may be statistically insignificant. Nevertheless, of all three compounds, only the OLG solution with UFBs may slightly increase the nucleation-promoting effect.

Considering now all three polyphenols and how their freezing behavior changed with UFBs, the results above suggest that UFBs contributed positively to OLG’s promotion of the ice nucleation probability, whereas UFBs had the opposite effect on TC’s behavior, with the nucleation slightly suppressed. In contrast, UFBs coexisting with TA led to no discernable change in the freezing temperature. These results suggest that different interactions may occur between these types of polyphenols and UFBs.

UFBs by themselves slightly promote nucleation, though not as much as that from each polyphenol. Any differences in the effects of these polyphenols on the nucleation activity of UFB-included water may arise from differences in the size and structure of polyphenols themselves or in their aggregate states in water. To help determine possible causes, additional experiments were run: observing the aggregate state of polyphenols in water, examining the change in the aggregate state in UFB-included water (or on the surface of an UFB), and measuring the size distributions of UFBs in water and in the polyphenol solution. Thus, we measured the size distribution of UFBs using the dynamic light scattering (DLS) method and observed the aggregate states of polyphenols and their interaction with UFBs using the FFT method.

### 3.4. Evaluation of the Mutual Effect of Polyphenols and UFB

For a further investigation of the influence of the polyphenols on the UFBs, we examined the solutions with the DLS and FFT methods. Solutions with polyphenols were all 0.1 wt%, consistent with the previous section. 

For the TA case, the UFB size distributions with and without TA show a similar double peak. The lower peak is near 250 nm, and the higher peak is above 1000 nm (Figure 9a). The close similarity indicates that the addition of TA does not change the size distribution of UFBs. The UFBs in this solution were generally found isolated as the examples in Figure 9b,c show, which also do not show any apparent differences to images of UFBs in pure water. These results suggest that TA and UFBs do not interact with each other in the aqueous solution, a result that seems consistent with the lack of any change that the UFBs caused to the TA freezing curve. Therefore, we suggest that TA is either completely dissolved in the aqueous solution or exists as small single aggregates in water, and also does not concentrate on the surface of UFBs. Under these conditions, the promotion effect of ice nucleation was mainly from the secondary nucleation phenomenon of TA, with little disturbance from the UFBs.

In contrast, when TC is added, the UFB distribution changes. Figure 10a shows the UFB size distribution changing from having twin peaks without the TC to having a single peak near a 300 nm diameter when the water has 0.1 wt% TC. Thus, by adding TC in the UFB-included water, the large UFBs apparently vanish, with only the small UFBs remaining. 

TEM images of the freeze-fractured replica of this aqueous solution indicate that fine particles deposited on the UFB surface (Figure 10b). Such a deposition is similar to the impurity accumulation found on UFBs in wastewater treatment [13], suggesting that aggregations of TC up to several tens of nm across may have formed while accumulating on the UFB interface. Such an accumulation may explain the change in the size distribution of UFBs as follows. UFBs with accumulated TC aggregates may tend to coalesce, becoming larger bubbles that rise to the sample top, thus exiting the bulk liquid phase. Such a process would also remove accumulated TC from the solution, thus decreasing the freeze-promoting effect of TC.

With added OLG, the UFB distribution also changes. In this case, it has three peaks: a small peak above 1300 nm, a large peak centered near 500 nm, and a middling peak below 200 nm (Figure 11a). As a reference, we also plotted the size distribution of small particles in 0.1 wt% OLG solution (without UFBs). These particles are considered to be aggregates of OLG, and have a peak size centered at approximately 250 nm. TEM images of the freeze-fractured replica of the UFB-included solution with OLG show that an impurity area exists around the UFB as two triangular “wing” features. An example is shown in Figure 11b. A thin layer was also found to form at the UFB interface (a “string” feature in the replica film). Therefore, it is likely that OLG in the aqueous solution also interacts with the UFBs, but differently than that with TC. Whereas the TC appears to form fine aggregates that accumulate on the UFB surface, the OLG seems to form larger aggregates that may cover the entire UFB. This difference in aggregation may arise because OLG has a larger molecular weight than TC.

To help to understand such an aggregation, we also show in the bar chart of Figure 11a the case with only a solution of 0.1 wt% of OLG (no UFBs). This case (open bars) shows a single peak centered around 250 nm that indicates an OLG aggregation phase. Such OLG aggregates may also form in the presence of UFBs, and if one such aggregate then attaches to a UFB (or instead aggregates on the UFB), it would form a compound aggregate. Such a process could then explain the largest peak in sizes near 500 nm for the UFB + OLG case (Figure 11a). A possible example of such a compound aggregation on a UFB is shown in Figure 11b. This image shows a thin layer covering the UFB plus two “wing” features—a layer that presumably consists of aggregated OLG. The coalescence of the compound aggregates may explain the larger-diameter peak above 1300 nm, whereas smaller OLG aggregates may explain the smaller-diameter peak below 200 nm. When such a distribution change occurs by the interaction between OLG and UFBs, the small UFBs may become more likely to collapse, and, in so doing, increase the probability of nucleating ice. As argued in the next section, this process may explain why the nucleation-promoting effect is largest for the OLG case. 

## 4. Discussion

The pure water samples froze in the range of −17 to −6 °C, which is slightly warmer than that observed in previous studies [4,5,6,7], presumably due to our larger sample volume. When we compared the freezing curve for pure water to that for UFB-included water (approximately 10^8^ mL^−1^ in UFB number concentration), we found them to be essentially the same. This indicates that any effect from the reactor wall and the surface UFBs are negligible in our experimental procedures. Still, compared to that of pure water, the freezing curve for UFB-included water shifted to a higher temperature by approximately 0.6 °C, which is just slightly larger than the temperature uncertainty. Thus, just as UFBs appear to promote gas-hydrate nucleation [15,16,17,18,19], the UFBs here may also promote ice nucleation. A possible promoting effect of UFBs may be caused by them providing a large area of the gas–liquid interface in water as potential nucleation sites. However, the effect is smaller than that observed for the gas-hydrate nucleation, which is already a small effect [15,16,17,18,19]. Nevertheless, the potential influences of UFBs on nucleation processes deserve further study.

Our focus was on the freezing curves of naturally occurring polyphenols (used by plants to acquire freeze tolerance) with and without UFBs. The three polyphenols were tannic acid (TA), tea catechin (TC), and oligonol (OLG), which previously had been found to be effective in suppressing the nucleation-promoting effect induced by ice nucleants such as AgI [6,7]. First, we measured their airborne ice-nucleation properties without any ice-nuclei additives in the present experimental setup, finding that they shifted the freezing curve by approximately 3 °C warmer compared to that of pure water. 

A previous study using droplets of approximately 2 μL for these same three compounds, all at a concentration of 0.1 wt%, reported differing effects on airborne ice nucleation [6]. They quantified the differences in nucleation behavior by measuring shifts in the FT50 values, defined as the temperature required for freezing 50% of the droplets. Specifically, TA and OLG had a slight supercooling-promoting activity with FT50 shifts of −1.7 ± 0.9 °C and −2.8 ± 1.3 °C, respectively, whereas TC had a slight nucleation-promoting effect with an FT50 shift of +5.3 ± 2.7 °C. The same group later measured the freezing temperature of TA, TC, and OLG solutions in much smaller droplets (10–30 μm diameter) using the W/O emulsion method, all at a concentration of approximately 0.1 wt% [7]. In some experiments, the droplets had AgI nucleant. They found the freezing curve of the polyphenol aqueous solutions without AgI nucleant to be almost the same as that of pure water, with neither nucleation-promoting nor nucleation-inhibiting effects being observed. Compared to the earlier airborne ice nucleation tests [6], however, we found that all three compounds had a nucleation-promoting effect, with a magnitude similar to that found only for TC.

In our experiments here, when UFBs were added to the TA solution, the freezing probability distribution was essentially unchanged; that is, there was no change in the freezing probability distribution with or without UFBs. This finding is consistent with the finding that the size distribution of the UFBs and the TEM observations of the UFBs were unchanged by adding TA to the solution. Therefore, we argue that the dissolved TA does not interact with UFBs. 

On the other hand, when UFBs were added to the TC solution, the freezing promotion effect was slightly smaller than with only TC; that is, the freezing curve shifted to a slightly lower temperature. Conversely, the opposite happened when UFBs were added to the OLG solution; that is, the freezing curve shifted slightly to higher temperature. The interaction of these polyphenols with UFBs was investigated using both DLS and FFT measurements. We found that different polyphenols had different aggregation states in water and different interactions with UFBs, resulting in changes to the UFB size distributions.

If the polyphenols accumulate on the UFB surface, the interfacial tension *σ* at the gas–liquid interface would likely change, assuming that the interfacial tension affects the bubble’s internal pressure and diameter according to the Young–Laplace equation [15]
(1)$$\Delta P=\frac{4\sigma }{d},$$
where Δ*P* is the internal bubble pressure and *d* is the bubble diameter. If *σ* is reduced by the accumulation of polyphenols and Δ*P* remains constant, *d* would decrease. This is consistent with the result of DLS measurements on TC and OLG experiments. Figure 11a shows that the average UFB diameter is approximately 670 nm without polyphenols, decreasing to approximately 346 and 400 nm when TC or OLG is added, respectively. Using Equation (1), we estimate that the modified interfacial tension *σ*’ due to the accumulation of TC or OLG is 0.52 or 0.60 times the surface tension σ of water. Such a decrease may have increased the probability of collapse for relatively small UFBs, and thus increased the probability of nucleating ice. If this is true, the difference in the effect of TC and OLG on the ice nucleation probability with UFBs would come from the coverage states of UFBs: the dissolved OLG seems to cover the entire UFB surface (Figure 11b), whereas TC accumulates on just parts of the UFB surface (Figure 10b). 

Although UFBs alone may not have a large promoting effect on ice nucleation, their physical properties, and hence nucleant properties, may be altered by adding a small amount of additives. Such effects may depend on the experimental procedure, particularly the sample volumes, cooling rates, and minor additives in the solution. In the future, we will search for and analyze effective additives that promote ice nucleation with UFBs, with the goal of developing control technology for water freezing that has a small environmental impact.

## 5. Conclusions

We investigated the nucleation of ice in UFB-suspended solutions. Compared to previous studies, we used a larger solution volume (1 mL) to reduce potential influences of the container wall on nucleation. The UFBs in the solution had a number density of 10^8^ mL^−1^. Based on multiple measurements of nucleation (more than 50 tests) under the same temperature, we could quantify changes to the nucleation activity by examining the shift in the freezing curve (i.e., the cumulative nucleation probability distribution). 

Compared to pure water, the freezing curves with the polyphenol solutions shifted approximately 2.7 °C higher with tannic acid (TA), approximately 3.5 °C higher with tea catechin (TC), and approximately 3.3 °C higher with oligonol (OLG). Then, when the UFBs were added to these solutions, additional temperature shifts in freezing curves occurred: slightly higher for OLG, essentially unchanged for TA, and slightly lower for TC. (UFBs without polyphenols only slightly promoted ice nucleation, shifting the pure-water freezing curve approximately 0.6 °C higher). 

To explain these differences, we measured the UFB size distributions using DLS and observed the UFBs in the freeze-fracture replicas using a TEM. These observations supported the freezing-curve shifts, indicating that OLG and TC modified the UFBs, whereas TA did not. We argued that different polyphenols had different aggregation states in water and different interactions with UFBs. 

If polyphenols accumulate on a UFB, they should affect some physical properties of the UFB, such as the surface tension and inner pressure. Such changes would affect the ice-nucleation activity of UFBs, as well as that of the polyphenols. In future studies, we plan to develop compound additives involving natural compounds and UFBs to promote ice nucleation. To reveal the mechanism of accelerating the ice nucleation by UFBs, and the interaction between UFB and additives in the solution more precisely, several experimental and theoretical investigations are required. These studies include the direct measurements of physical properties of UFBs and the stability change in UFBs with additives, and systematic measurements of ice-nucleating activities of various polyphenols with UFBs.

## Figures and Tables

**Figure 1 nanomaterials-13-00205-f001:**
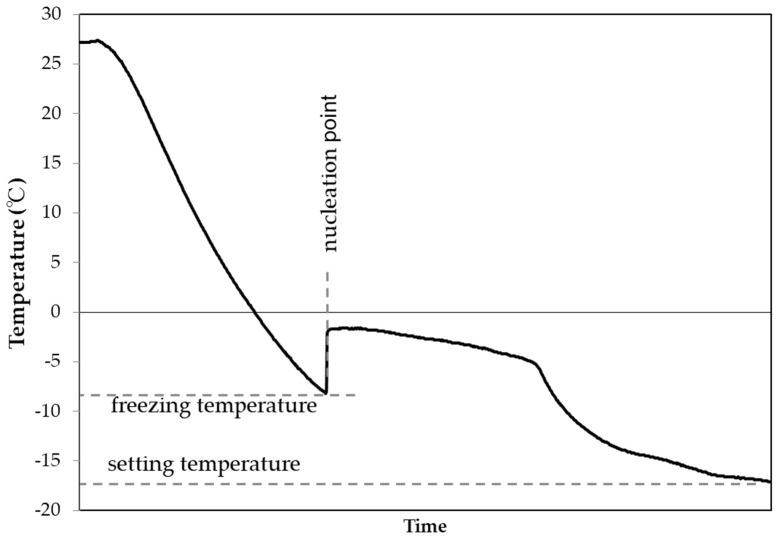
Typical sample temperature profile from room temperature to freezing.

**Figure 2 nanomaterials-13-00205-f002:**
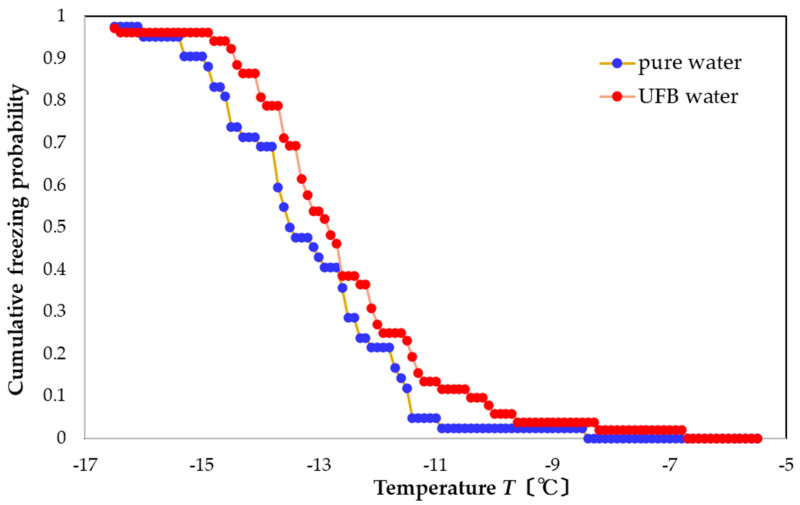
Freezing curves of pure water and UFB water, one day after preparation.

**Figure 3 nanomaterials-13-00205-f003:**
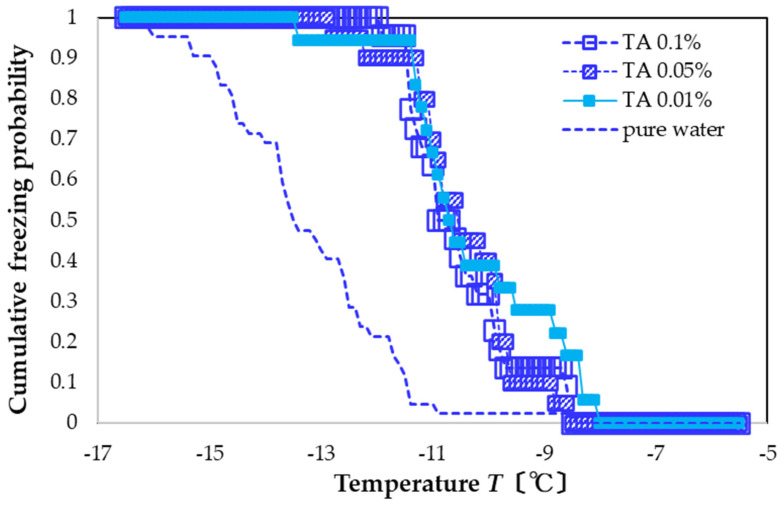
Freezing curves of TA solutions. Leftmost blue-dashed curve is the freezing curve of pure water.

**Figure 4 nanomaterials-13-00205-f004:**
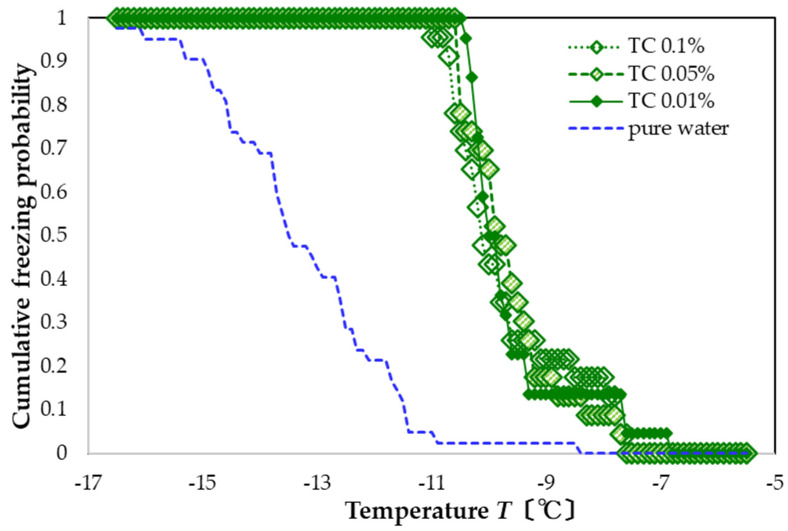
Freezing curves of TC solutions. Leftmost blue-dashed curve is the freezing curve of pure water.

**Figure 5 nanomaterials-13-00205-f005:**
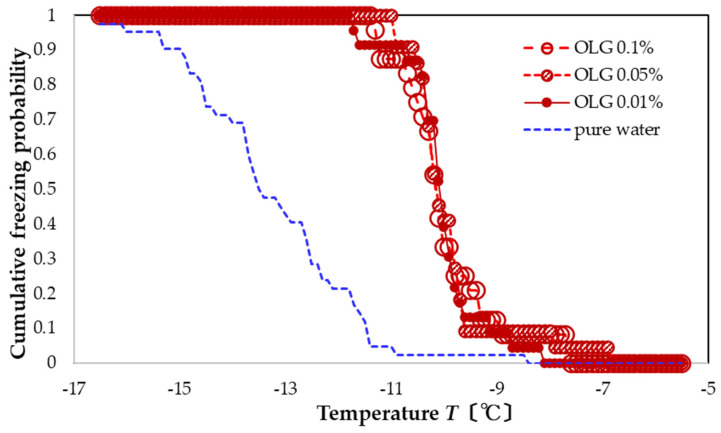
Freezing curves of OLG solutions. Leftmost blue-dashed curve is the freezing curve of pure water.

**Figure 6 nanomaterials-13-00205-f006:**
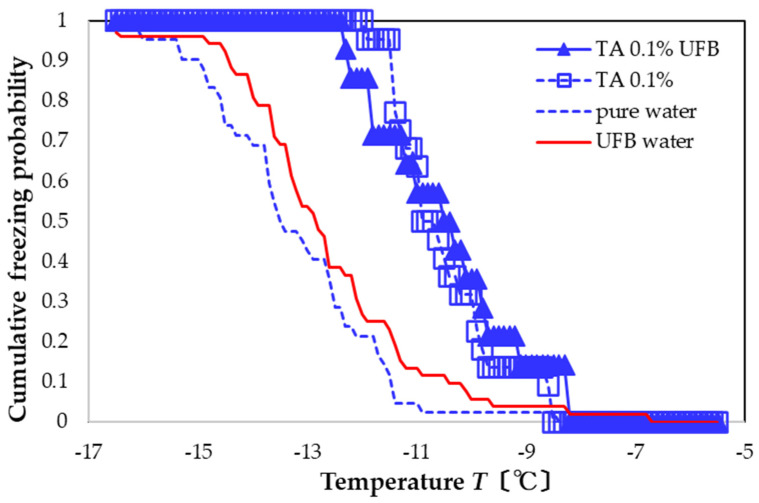
Freezing curve of aqueous solution with 0.1 wt% TA dissolved in water with UFBs (solid blue). Other curves shown for comparison (same as Figure 2).

**Figure 7 nanomaterials-13-00205-f007:**
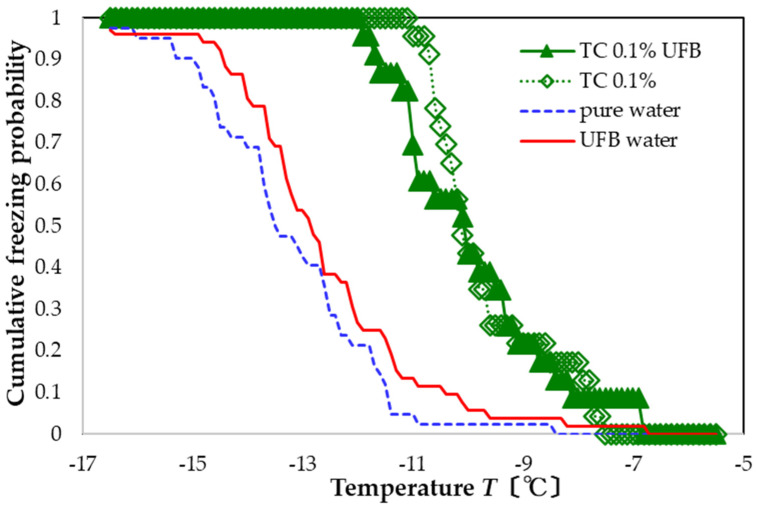
Freezing curve of aqueous solution with 0.1 wt% TC dissolved in water with UFBs (solid green). Other curves shown for comparison.

**Figure 8 nanomaterials-13-00205-f008:**
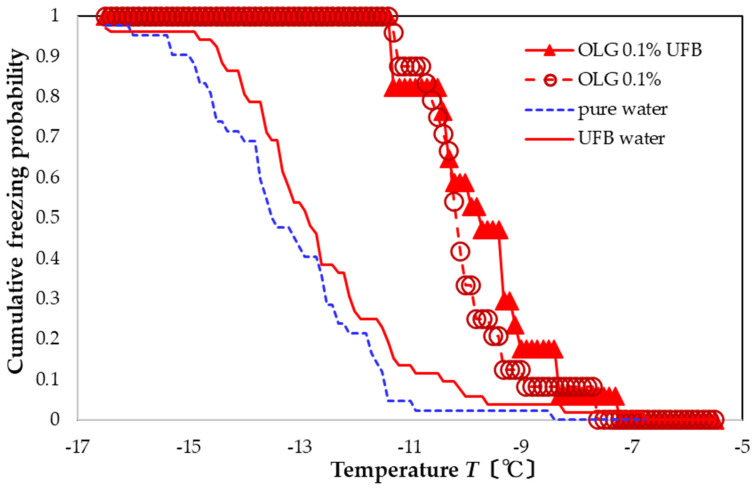
Freezing curve of aqueous solution with 0.1 wt% OLG dissolved in water with UFBs (solid red). Other curves shown for comparison.

**Figure 9 nanomaterials-13-00205-f009:**
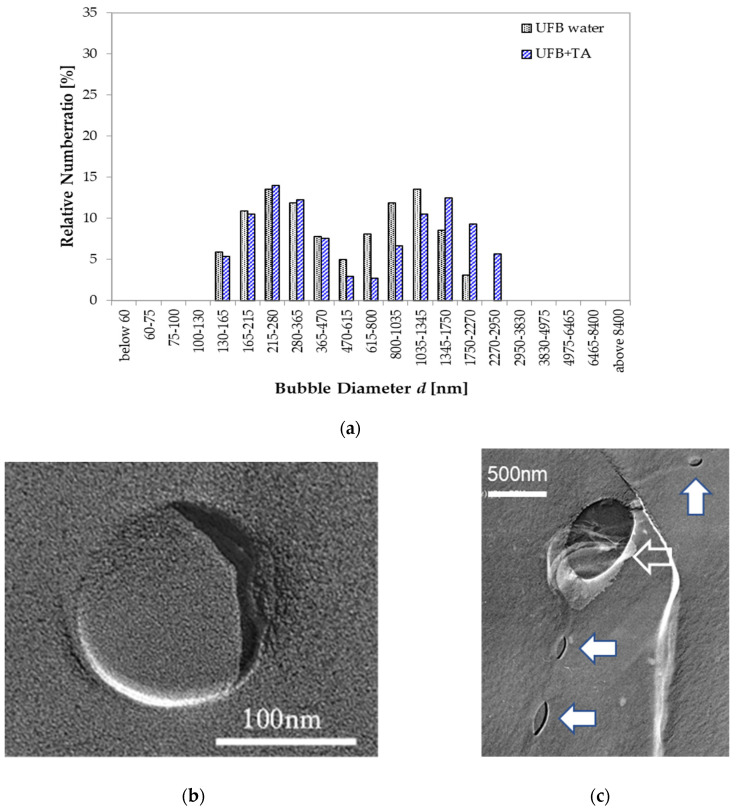
UFB sizes with TA. (**a**) Size distributions in pure water (gray solid) and in water with TA of 0.1 wt% (blue hatched). Ordinate scale is the fraction of UFBs per listed size range (both distributions add up to 100%). (**b**) TEM image of freeze-fracture replica of a UFB in the 0.1 wt% TA solution observed at high magnification. (**c**) Same as (**b**) except different UFBs and a lower magnification. Both smaller UFBs (solid allows) and a larger UFB (open allow) exist on an ice grain boundary.

**Figure 10 nanomaterials-13-00205-f010:**
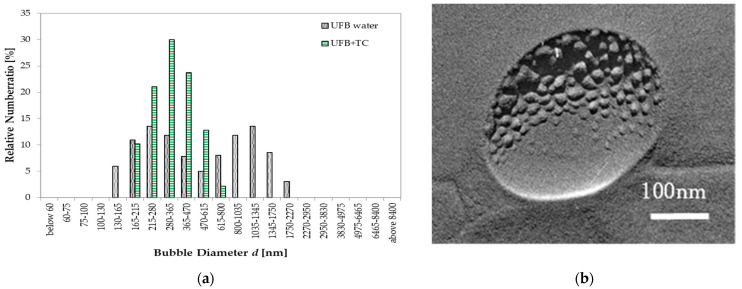
UFB sizes with TC. (**a**) Size distributions in pure water (gray solid) and in water with TC of 0.1 wt% (green hatched). Ordinate scale is the fraction of UFBs per listed size range (both distributions add up to 100%). (**b**) TEM image of freeze-fracture replica of a UFB in the TC solution.

**Figure 11 nanomaterials-13-00205-f011:**
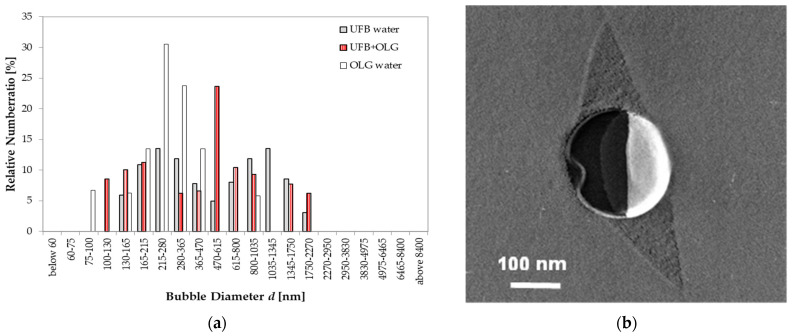
UFB sizes with OLG. (**a**) Size distributions in pure water (gray solid) and in water with OLG of 0.1 wt% (red hatched). Ordinate scale is the fraction of UFBs per listed size range (both distributions add up to 100%). Also shown in (**a**) are the particle sizes in 0.1 wt% OLG solution (open bar). (**b**) TEM image of freeze-fracture replica of a UFB in the OLG solution.

## Data Availability

The numerical data presented in this study are available in Appendix A.

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
