# Peer review of "Effect of Polyphenols on the Ice-Nucleation Activity of Ultrafine Bubbles"

_nanomaterials, 2023, doi:10.3390/nano13010205_

Round 1
Reviewer 1 Report
This manuscript presents a study on the effect of Ultrafine bubbles (UFBs) and polyphenols on the promotion or suppresion of ice crystals nucleation. The manuscript is generally well written and and well structured. However, some specific issues should be revised, changed and corrected and I hereby give my detailed comments to improve this manuscript.
1. keywords need to be revised avoiding long phrases and be more concise
2. some choice of vocabulary need to be revised, e.g. page 1 line 33"stably", page 1 line 43 "Hence"
3. Some statments within the intrduction and discussion lack cited references, e.g sentences on line44-46 lacks at least two references.
4. line 71: please specify how long is this "while" ?
5. One factor that should affect the stability of the generted UFB is the surface tension that may be different for bubles of difference sizes. The authors are talking about a size variation from 1mm down to 100nm and this factor should be discussed thoroughly.
6. Vetrification, as an effective method to avoid the formation of ice crystals was not discussed anywhere and, in my opinion, should have had the authors' attention.
7. The figure captions need to be fully descriptive and contain all the information, even if they only introduce one difference with a preceding figure.
8. More importanatly, since the temperature profile figures shows temperature variations from around -15 to -5 celsius degrees, the authors should expand the figures to show clearly the differences in the temperature profiles at different concentrations of the polyphenols, besides they should use different coloring for every concentration.
9. In the experimental section, the prepartion of the samples for TEM inspection was not well described and should be indicated, since in such cases ice crystals can be formed durin the vetrification process itself.
10. Addittionally, in the results section, TEM imaging should be more representative. Images at low magnification showing the general trend followed by a magnified image for one UFB should replace the currently present figures in this manuscripts.
11. Introducing very many UFBs in a specific solution implies the presence of a lot of gas-liquid interfaces, that should promote the vapor diffusion and nucleation. How this can be discussed within the context of this study?
12. Finally, the manuscript lacks a conclusion section that I think should summarize the study findings and implications for future research.
Reviewer 2 Report
In daily production and life, controlling freezing temperature is an important means to maintain the normal growth and survival of organisms. Ultrafine bubbles in water are important substances to control the freezing of water, and adding different additives to water is another important method to control the freezing temperature. The influence of ultrafine bubbles and other three different additives, TA, TC, and OLG, on water freezing nucleation was analyzed under controlled experiment conditions. Different from the previous experiments, the authors used a solution with a larger sample and obtained different results from the previous ones, especially for TC and OLG additions. At the same time, the radius distribution characteristics of bubbles in different mixing states were obtained by dynamic light scattering method, and the interaction of different mixtures with pure water and ultrafine bubbles was analyzed.
The paper has some innovation, the experimental data and results are credible, and the language expression is clear. But there are still a few questions that the authors need to clarify:
1. The accuracy of temperature measurement is ±0.5℃, which is too low to accurately mark the temperature change process of ice nucleation condensation in the laboratory. For example, when the FAC is 0.3℃, it cannot be explained whether it is the measurement error or the real effect of temperature change. If the experimental conditions permit, it is suggested that the authors use a more refined temperature measuring device for analysis. In addition, please explain the measurement frequency of the temperature data logger.
2. The description of the radius distribution characteristics of UFB in figure 9 and figure 11 is not accurate. When TA is not added in FIG. 9, the higher peak is around 1200nm (not 1000nm descripted in the text). However, in Figure 11, the radius description of the three peaks is not accurate, and there is no peak at 1300nm.
3. In addition, in Figure 11, there is no OLG exist in the range of 1035-1345nm. Why? Please show the reason.
Round 2
Reviewer 1 Report
The authors have addressed most of my suggested modifications and recommendations. the colours in the temperature profiles were changed, yet they did not understand the aim of the change... I leave this for the final editing of the manuscript. As it is now, I am happy to recommend its publication
Reviewer 2 Report
Thanks to the authors for their serious revision. The revised article has a more compact structure, more accurate language, and is easier to understand by readers. The authors' replies answer my doubts, and I feel that the revised article satisfies the publication requirements of the journal and is recommended to be accepted for publication.